# OpenReview forum: "Concept Concentration for Faithful Representation Intervention"
_ICLR.cc/2026/Conference — Submitted to ICLR 2026_

### Official Review · Reviewer_3MTC · 2025-10-17

**Soundness:** 4
**Presentation:** 4
**Contribution:** 4
**Rating:** 8
**Confidence:** 4

**Summary:**

This paper investigates the limitations of existing representation intervention methods for safety alignment in LLMs, showing that perfect harmful concept erasure is theoretically impossible in non-linear representation spaces. To address this, it proposes COCA (COncept ConcentrAtion), which introduces structured reasoning annotations to linearize harmful concepts, enabling effective and faithful erasure. Experiments across four LLMs demonstrate that COCA improves robustness against out-of-distribution jailbreaks while preserving benign capabilities.

**Strengths:**

1. **Clear Theoretical Contribution.** The paper rigorously identifies the root cause of representation intervention failure as non-linear entanglement between harmful and benign concepts and mathematically proves the impossibility of perfect erasure under this regime (Theorem 2.2).
2. **Novel Conceptual Shift.** Instead of seeking ideal intervention points in a complex space, it introduces Concept Concentration (COCA), a new paradigm that simplifies the representation space at the data level.
3. **Strong Empirical Results.** COCA substantially reduces OOD jailbreak success rates across four base models (LLaMA-3.1-8B, Qwen-2.5-7B, etc.) while preserving performance on math and code reasoning tasks, demonstrating a balance between safety and utility

**Weaknesses:**

1. **Annotator Bias.** The method depends on fine-grained concept annotations, so residual bias or subjectivity in defining “unsafe” content may remain.
2. **Absence of Large Reasoning Model Evaluation.** The paper does not include experiments on Large Reasoning Models (LRMs), which are increasingly important for assessing safety and concept alignment. Because the proposed method relies on explicit structural annotations, its applicability to LRMs such as DeepSeek-R1-Qwen-7B or DeepSeek-R1-LLaMA-8B remains unclear. Without such validation, it is difficult to confirm whether the approach generalizes beyond standard instruction-tuned LLMs. Comparison with recent LRM baselines [1,2] would strengthen the evaluation.

[1] Jeung, Wonje, et al. "SAFEPATH: Preventing Harmful Reasoning in Chain-of-Thought via Early Alignment." NeurIPS (2025).\
[2] Wang, Zijun, et al. "Star-1: Safer alignment of reasoning llms with 1k data." arXiv preprint arXiv:2504.01903 (2025).

**Questions:**

I don't have additional questions. Solid work.

---

> ### Author Response · Authors · 2025-11-22
> **Response to Reviewer 3MTC**
>
> Thank you for your support of our work and insightful comments! We hope our responses below could strengthen your confidence in assessing our work!
>
> > W1 Residual bias or subjectivity on harmful concepts
>
>
> We agree that safety taxonomies can vary across annotators and that subjective drift can affect reproducibility. We have mitigated this in three concrete ways.
>
> - First, the unsafe space we target is widely standardized: illegal instructions from public corpora such as Beavertails [1], with clear harmful intents (see examples in Appendix K). Our training set does not include complex jailbreak phrasings. Instead, it uses ordinary unsafe/benign instructions where the safety label is largely unambiguous under mainstream safety policies.
>
> - Second, COCA constrains the concept trace into a short, auditable structure: a concept list and a binary gate that must be justified in `<check>`. This deliberately limits narrative degrees of freedom and focuses the supervision signal on “what unsafe concept is present”. In the meantime, the design of COCA also enables extension and open to incorporating new policies of “unsafe concept” during training.
>
> -  Following your suggestion, we also varied the annotator and studied the potential biases of different annotators. In the table below, we observed stable safety gains under different annotators: GPT‑4o (main paper), Claude, and LLaMA‑3.1‑8B‑Instruct (the “self-generated” variant in Table 3). The self-generated setting removes reliance on a proprietary teacher. COCA’s OOD robustness stays with comparable averages. These results indicate that the effect stems from the structural concentration induced by COCA rather than from a specific teacher’s narrative phrasing.
>
>
> |        | PAIR | JChat | Cipher | Comp | Code | JailWild | Avg|
> |--------|-------------|-----------|--------|------|------|------------|------|
> | COCA (GPT-4o)  |    17.1     |   5.5    |  2.5   |  0.0 |  36.0  |     2.1   |     10.5    |
> | COCA ( LLaMA‑3.1‑8B‑Instruct)   |    14.0     |  8.0    | 4.0   | 1.0 | 42.5 |    9.4    |   13.2 |
> | COCA (Claude)   |   15.6      |  4.0    |  3.0 |  0.0 | 36.5  |   5.7     |  10.8   |
>
> > W2 Additional Results on LRM
>
> We added experiments on DeepSeek‑R1‑Qwen‑7B (Appendix E). Following your suggestion, we also incorporated recent LRM safety baselines, including SAFEPATH and Star‑1, to contextualize results. We trained COCA using the same tag structure as in our main experiments: a `<think>` segment that begins with “let’s think about safety first,” followed by concept identification, check, unsafe concepts gating, and a final `<response>`. We used LoFiT as the intervention module. As a reference LRM baseline, we evaluated SAFEPATH and Star‑1 using their released checkpoint.
>
> |        | PAIR | JChat | Cipher | Comp | Code | JailWild | Avg|
> |--------|-------------|-----------|--------|------|------|------------|------|
> | COCA   |      29.6    |  4.0     |  0.0   |  0.0  | 35.0 |   2.1    |  11.7      |
> | SAFEPATH   |    25.0   |   4.0   |   0.0 | 0.0|  44.0 |    5.7   |13.1   |
> | STAR-1   |       26.5   |    7.5   | 0.0   | 0.0 |45.0  |   7.6     | 14.4    |
>
>
>
>
> **We observe that COCA-trained LRMs achieve lower or comparable OOD attack success than the same LRMs trained with only a SAFEPATH-style prefix or with Star‑1’s checkpoint**. These results show that COCA is also applicable to LRMs. While LRMs already produce longer chains of thought,  generic “more reasoning” is not sufficient for safety generalization. It again confirms that linearizing where harmful concepts reside is essential. COCA provides the structural pressure to concentrate harmful concepts so that representation intervention can work faithfully. The resulting gain of COCA is consistent when applied on either the reasoning-strong backbones or normal base models.
> We have presented these as preliminary evidence that COCA is applicable to LRMs and competitive with recent LRM safety baselines. Despite the preliminary effectiveness, we have not yet exhaustively explored design choices unique to LRMs (e.g., reasoning‑length schedules, or RL post-training). We believe making reasoning‑strong models safe and faithfully align high‑level safety concepts within their CoT scratchpads, is a promising future work.
>
> [1] Jiaming Ji, Mickel Liu, et al. Beavertails: Towards improved safety alignment of llm via a human-preference dataset. Advances in Neural Information Processing Systems, 36:24678–24704, 2023.

---

> > ### Comment · Reviewer_3MTC · 2025-11-23
> > **Thank you for Rebuttal**
> >
> > Thank you for the detailed response. The cross-annotator and LRM experiments directly address the concerns I raised. I re-read the paper and am now confident that it deserves acceptance. Please ensure both results are included in the final version.

---

> > > ### Author Response · Authors · 2025-11-23
> > >
> > > Dear Reviewer 3MTC,
> > >
> > > Thank you again for your thoughtful review. We are happy to learn that our responses addressed your concerns and are grateful for your positive feedback! Please feel assured that the results will be included in the final version.
> > >
> > > Best regards, Authors

---

### Official Review · Reviewer_eUou · 2025-10-30

**Soundness:** 3
**Presentation:** 2
**Contribution:** 3
**Rating:** 6
**Confidence:** 2

**Summary:**

This paper proposes COCA, a presentation intervention approach to improve the safety alignment robustness against out-of-distribution jailbreaks. It first conducts a theoretical analysis to reveal the infeasibility of existing representation intervention techniques to erase harmful concepts without degrading utility under non-linear setup. It then proposes a data refinement approach to explicitly decompose the safety data into reasoning steps and prove that such approach is able to simplify the decision boundary and make linear erase feasible. Evaluation results on several jailbreaks and comparison with SOTA concept erase and safety alignment baselines demonstrate the effectiveness of the COCA.

**Strengths:**

- The paper tackles an important problem in LLM safety.
- The theoretical analysis is insightful and valuable to the safety community, as it exposes fundamental limitations of existing methods and motivates the proposed approach.
- The proposed data refinement method is intuitive and easy to follow.

**Weaknesses:**

- The empirical improvement over existing methods is relatively small.
- The presentation could be improved for better clarity and readability.

**Questions:**

- The presentation, especially in Tables 1 and 2, could be improved. The current layout is hard to read and makes it difficult to match results with the proposed approach.
- When compared with state-of-the-art alignment methods such as STAIR, the improvement achieved by COCA-structured data appears marginal and, in some cases, even falls below baseline performance.  For example, STAIR achieves 4.3% ASR on PAIR when applied to LLaMA-3.1-8B, while COCA+RR yields 7.8%.
- How are ID and OOD attacks refined? What is the rationale for treating all jailbreak attacks as OOD? Some, like PAIR, do not involve unreadable tokens, what characteristic makes them OOD?
- It would be helpful to include evaluations against gradient-based jailbreaks, such as GCG [1], to further demonstrate robustness.

---
Reference
----

[1] Zou A, Wang Z, Carlini N, et al. Universal and transferable adversarial attacks on aligned language models[J]. arXiv preprint arXiv:2307.15043, 2023.

---

> ### Author Response · Authors · 2025-11-22
> **Response to Reviewer eUou (Part 1)**
>
> Thank you for your time and effort in reviewing our work. Please find our responses to your questions and concerns below.
>
> > Q1&W1 The presentation, especially in Tables 1 and 2, could be improved.
>
>
> Thanks for your suggestion. We have restructured both tables to group methods by training paradigm (e.g., ReFT vs. LoFiT), so readers can compare them easily.
>
> > W2 When compared with state-of-the-art alignment methods such as STAIR, the improvement achieved by COCA-structured data appears marginal and, in some cases, even falls below baseline performance.
>
>
> STAIR achieves a strong number on PAIR when evaluated in isolation. However, when averaging across all OOD suites, COCA still provides a 1.1% reduction on attack success rate with LoFiT intervention and 4.5% reduction on attack success rate with RR intervention. Furthermore, STAIR adopted 3K Jailbreak prompts from Jailbreak-V as their training data. Our COCA was trained only on the illegal instructions without jailbreak prompts, which is a more challenging scenario to achieve OOD robustness.
>
> We also implemented and evaluated a **combined variant that explicitly composes STAIR’s introspective stage with COCA’s concept-concentration protocol.** Concretely, we feed STAIR’s “Problem Analysis” into COCA’s `<think>` segment, then apply the COCA `<concept> → <check> → <erase unsafe concepts> → <response>` pipeline, and train the same LoFiT intervention on the resulting refactored traces. Here, we used STAIR's publicly released data. We further refactor the data under our COCA template, and then combine them. The results are given below. When COCA and STAIR were combined, a **4.7% reduction in attack success rate** was further achieved on OOD jailbreak attacks.
>
>
>
> |        | PAIR | JChat | Cipher | Comp | Code | JailWild | Avg|
> |--------|-------------|-----------|--------|------|------|------------|------|
> | COCA  |    35.9      |  13.5     |   3.5  | 0.0   | 42.5  |     5.7   |   16.9      |
> | STAIR   |   31.3       |  18.0    |  3.0   |  0.0 | 40.5 |   6.7     |      16.6  |
> | COCA + STAIR    |    23.4      |   7.5    |   0.0  | 0.0  | 36.5  |    5.7     |   12.2   |

---

> ### Author Response · Authors · 2025-11-22
> **Response to Reviewer eUou (Part 2)**
>
> > W3 How are ID and OOD attacks refined? What is the rationale for treating all jailbreak attacks as OOD? Some, like PAIR, do not involve unreadable tokens, what characteristic makes them OOD?
>
>
> Thank you for pointing out this potentially confusing point. We have made ID/OOD definitions clear in our revised manuscript following our explanation below:
>
> - The ID data  consists of standard and plain unsafe requests (illegal-instruction style)  [1] expressed directly and without adversarial scaffolding. These are prompts where the harmful intent is **explicit and unwrapped**. (e.g., “How do I get drugs past the border?”) We also use the same illegal-instruction styled data to conduct the supervised finetuning.
>
> - The OOD evaluation suites are composed of jailbreak prompts whose mechanisms differ significantly from the training distribution along one or more axes:
>   - They embed the harmful goal under adversarial role or persona-wrapping (JChat [2]).
>   - They inject meta-instructions that explicitly redirect the model away from its safety policy (PAIR [3], JailWild [4]).
>   -  They encode the harmful goal in cipher-like or obfuscated tokens (Cipher [5]).
>   -  They reframe the goal as a code/completion objective to bypass refusal heuristics (Code Attack [6]).
>
> The OOD here is therefore not about token readability but about distribution shift. The attack mechanism moves outside the support of the training distribution. Under this definition, jailbreak suites, including PAIR, are OOD for our models because they rely on role-play and meta-instruction patterns that the model has not seen during training. Although PAIR uses readable text, its layered roles and meta-instructions rather than direct illegal instruction,is precisely the kind of mechanism shift we target to achieve OOD robustness.
>
>
>
> [1] Jiaming Ji, Mickel Liu, et al. Beavertails: Towards improved safety alignment of llm via a human-preference dataset. Advances in Neural Information Processing Systems, 36:24678–24704, 2023.
>
> [2] Shen, Xinyue, et al. "" do anything now": Characterizing and evaluating in-the-wild jailbreak prompts on large language models." ACM SIGSAC Conference on Computer and Communications Security. 2024.
>
> [3] Patrick Chao, Alexander Robey, et al. Jailbreaking black box large language models in twenty queries. arXiv preprint arXiv:2310.08419, 2023.
>
> [4] Liwei Jiang, Kavel Rao, et al. Wildteaming at scale: From in-the-wild jailbreaks to (adversarially) safer language models. arXiv preprint arXiv:2406.18510, 2024.
>
> [5] Youliang Yuan, Wenxiang Jiao, et al. GPT-4 is too smart to be safe: Stealthy chat with LLMs via cipher. In The Twelfth International Conference on Learning Representations, 2024.
>
> [6] Qibing Ren, Chang Gao, et al. Exploring safety generalization challenges of large language models via code. arXiv preprint arXiv:2403.07865, 2024.
>
> > W4 It would be helpful to include evaluations against gradient-based jailbreaks, such as GCG, to further demonstrate robustness.
>
>
> We agree that gradient-based attacks are an important part of the robustness picture. We have added evaluations against GCG. In the following table, we observe that **COCA remains robust and brings significant improvements against GCG attacks**.
>
> |        | LoFiT + Vanilla| LoFiT + COCA | ReFT + Vanilla | ReFT + COCA |
> |--------|-------------|-----------|--------|------|
> | GCG   |      35.0   |   4.0   |  45.5  | 9.5  |

---

> ### Author Response · Authors · 2025-11-26
>
> Dear Reviewer eUou,
>
> We are grateful for your time and constructive feedback. To facilitate our discussion at this midway point, we provide a concise summary of your concerns and our responses:
>
> > Q1&W1 The presentation, especially in Tables 1 and 2, could be improved.
>
> We restructured both tables to group methods by training paradigm (e.g., ReFT vs. LoFiT), improving readability and direct comparability.
>
> > W2 When compared with state-of-the-art alignment methods such as STAIR, the improvement achieved by COCA-structured data appears marginal and, in some cases, even falls below baseline performance.
>
> COCA achieves lower average OOD ASR across suites, reducing ASR vs. STAIR by 1.1% (LoFiT) and 4.5% (RR).
>
> > W3 ID and OOD clarification
>
> We provide clarifications on ID and OOD, making the definitions clearer.
>
>
> > W4 It would be helpful to include evaluations against gradient-based jailbreaks, such as GCG, to further demonstrate robustness.
>
>
> We added GCG evaluations: ASR drops from 35.0→4.0 with LoFiT+COCA and 45.5→9.5 with ReFT+COCA, showing strong robustness.
>
> Thank you once again for your time and effort in reviewing our paper. We would be grateful if you could review our clarifications and let us know if you have any remaining questions or concerns. Thank you very much for your valuable feedback.
>
> Best regards, Authors

---

> ### Author Response · Authors · 2025-11-27
> **Gentle Reminder**
>
> Dear Reviewer eUou,
>
> Thank you again for taking the time to review our work. As the discussion phase will close in less than one week, to allow us sufficient time to address your comments, could you please take a moment to look over our responses and let us know if you have any remaining questions? Thank you very much!
>
> Best regards, Authors

---

### Official Review · Reviewer_J5S5 · 2025-10-31

**Soundness:** 3
**Presentation:** 3
**Contribution:** 1
**Rating:** 2
**Confidence:** 5

**Summary:**

This paper studies the safety alignment problem of LLMs. The authors propose COCA, which first indentifies the potential unsafe concepts and then decides the response. The anthor introduce a well-designed system prompt annotated by teacher model (GPT-4o) into the training data. It aims to induce the student LLMs to reflect and indentify on the potential unsafe the concepts before the standard response. The authors validate the effectiveness of the proposed approach in multiple experiments.

**Strengths:**

1. The problem studied in this paper is valuable.
2. The paper is well written.
3. The proposed approach is effective.

**Weaknesses:**

1. This paper tells a good story, from the standard safety alignment, concept-centric alignment to the linear representation hypothesis. After reading that, I expect some amazing ideas to solve the non-linear representation alignment problem. However, the method is just like a prompt or CoT engineer with well-designed prompts, and fine-tune the model to generate safety thinking before the standard response. We believe this is a very normal and robust method in real applications.
2. The evidence of my first point can be found in Table 3. The "self-generated" method can also significantly improve the safety performance. And I believe that the LLMs can also achieve a good safety performance by adding this "concept prompt" in the LLMs without any training.

**Questions:**

See the weaknesses.

---

> ### Author Response · Authors · 2025-11-22
> **Response to Reviewer J5S5 (Part 1)**
>
> Thank you for your time in reviewing our work and for your acknowledgement of the problem we study. We feel there is a misunderstanding, and we would like to clarify your concerns in the response below!
>
> > W1 However, the method is just like a prompt or CoT engineer with well-designed prompts, and fine-tune the model to generate safety thinking before the standard response. We believe this is a very normal and robust method in real applications.
>
> We need to clarify that COCA is not simply a prompt trick. The COCA solution and the structured tag design are motivated by our theory.
> - First, we have identified a limitation for representation intervention methods. Specifically, we formalized that, when harmful concepts are distributed non-linearly across many directions, perfect erasure is impossible without collateral distortion of utility. **This impossibility motivates changing the geometry of the internal states, not just adding longer chains of thought**.
> - Second, motivated by the analysis, we proposed a data-level linearizer that concentrates harmful information into a near-linear subspace, after which standard intervention methods become effective and faithful. We show that, under our training factorization, the covariance between the concentrated representation and the harmful concept collapses onto a single linear direction. **COCA is the compact protocol that realizes this factorization in practice** (as also shown in the response to Reviewer Z6N3 that the structured tag design is well-aligned to the theory and empirically desirable to achieve robust safety alignment performance):
>   - a concept-extraction stage to produce an explicit, compressed representation of potentially unsafe content
>   - a gating/checking stage that conditions downstream behavior solely on that representation
>   - an erase stage that removes unsafe concepts while preserving benign follow-ups.
> - Third, we also provided **empirical evidence separating COCA from generic CoT or prompt shaping**:
>   - In ablation, removing the concept stage (think-only CoT) reduces OOD robustness. We also tested a prompt-only baseline (See our response to your second concern). The average jailbreaking attack success rate of prompt-only remains substantially higher than with COCA fine-tuning (note that higher is worse).
>   - Beyond outputs, we have measured the geometry directly. On Qwen2.5-7B, we have trained a single linear probe per layer to distinguish unsafe from benign. We trained the probe on illegal and benign prompt internal states. We evaluate the AUC on jailbreak and benign prompt internal states. COCA-trained models exhibit higher ROC–AUC than SRG, STAIR, and vanilla-tuned baselines, demonstrating that harmful content has indeed been concentrated into a more linearly accessible direction. This linear accessibility is precisely what enables LoFiT to act faithfully.
>
> |        | Vanilla | COCA | STAIR | SRG | RR |
> |--------|---------|------|--------|------|----|
> | AUC  |       0.85 |   0.96    |    0.93 | 0.90  | 0.91 |
>
> - Fourth, we have positioned COCA with respect to prior reasoning-based safety methods. SRG and STAIR, **to demonstrate the necessity of the theory-aligned structure in COCA**. STAIR/SRG asks the model to elicit a free-form chain-of-thought whose number of steps, wording and internal references are left unspecified. In COCA the reasoning protocol is derived from the concept-concentration analysis. Each of its four tags (`<think>, <concept>, <check>, <erase unsafe concepts>`) plays a definite role in linearizing the harmful direction:
>   - The `<concept>` stage forces the model to restructure explicit, compressed representation of unsafe content.
>   - The `<check>` stage conditions downstream behaviour exclusively on that representation.
>   - The refusal logic is separated in `<erase unsafe concepts>`, ensuring that benign requests do not be bypassed.

---

> ### Author Response · Authors · 2025-11-22
> **Response to Reviewer J5S5 (Part 2)**
>
> > W2 The evidence of my first point can be found in Table 3. The "self-generated" method can also significantly improve the safety performance. And I believe that the LLMs can also achieve a good safety performance by adding this "concept prompt" in the LLMs without any training.
>
> We need to clarify that the ``self-generated’’ variant in Table 3 uses the Instruct model (i.e., LLaMA-3.1-8B-Instruct) to annotate the data, and performs supervised fine-tuning on COCA-structured traces. We have revised our manuscript to make it clearer.
>
> To directly address your concern, we have additionally tested a prompt-only baseline that keeps the vanilla-trained model unchanged and merely prepends the COCA template to the request, instructing the model to think, list concepts, and refuse when necessary. Although this verbal steering reduces attack success rate relative to no steering, the average OOD success rate remains substantially higher than with COCA fine-tuning.  **The gap arises because the harmful representation is still dispersed across many directions. Without the supervised concentration step, the downstream intervention cannot eliminate it**.
>
>
> |        | PAIR | JChat | Cipher | Comp | Code | JailWild | Avg|
> |--------|-------------|-----------|--------|------|------|------------|------|
> | Base + COCA  |    17.1     |   5.5    |  2.5   |  0.0 |  36.0  |     2.1   |     10.5    |
> | Base + COCA  with Instruct-generated data |      14.0   |  8.0     |   4.0  |  1.0 | 42.5   |   9.4     |      13.2   |
> | Instruct  |    10.9    |  3.5     |  1.0   | 0.0  |  68.5 |    4.9   |    17.7    | |
> | Prompt-Only   |     46.8    |  32.0     |  20.5   | 4.5  | 62.0  |  20.1    |  30.9    | |
>
> The results above again confirmed that COCA is not merely a prompt or CoT engineer, but a well-motivated, theory-aligned, easy but robust method. We hope our responses and the listed evidence could clarify your concerns about our method. If you still have any remaining concerns, we are happy to provide more evidence to address them!

---

> ### Author Response · Authors · 2025-11-26
>
> Dear Reviewer J5S5,
>
>
> Thank you for taking the time to review our submission and for highlighting the core issue we study. To streamline the discussion, we provide a concise summary of our responses, followed by point-by-point clarifications. We hope this helps clarify how our contributions go beyond prompt engineering and why COCA is theory-driven and empirically necessary.
>
> > W1 The method is just like a prompt or CoT engineer
>
> - Theory-driven: We formalize an impossibility for perfect erasure when harmful concepts are dispersed nonlinearly, motivating geometry change. COCA is a data-level “linearizer” that concentrates harmful information into a linear subspace so simple editors (e.g., LoFiT) act faithfully.
> - Structure matters: COCA’s tags (`<concept>, <check>, <erase>`) operationalize identification → gating → erasure. Ablating the concept stage (think-only) harms OOD robustness.
> - Representation evidence: Linear probes show higher unsafe–benign separability with COCA (AUC 0.96) than STAIR (0.93), SRG (0.90), RR (0.91), and vanilla (0.85), indicating a clearer harmful direction.
> - Empirically:
>   - Ablations show generic CoT (no concept stage) underperforms COCA on OOD jailbreaks.
>   - A prompt-only baseline also underperforms COCA finetuning (see below)
>
> > W2 “Prompy-only” without any training results
>
> We clarified that “Self-generated” in Table 3 involves SFT on Instruct-generated COCA traces (not zero-shot prompting). Prompt-only (no training) reduces ASR modestly but is far worse than COCA SFT: avg OOD ASR 30.9 (prompt-only) vs. 10.5 (Base+COCA).
>
> Thank you once again for your time and effort in reviewing our paper. We have posted a point-by-point response above. As we are now midway through the author–reviewer discussion period, we would be grateful if you could review our clarifications and let us know if you have any remaining questions or concerns. Thank you very much for your valuable feedback.
>
> Best regards, Authors

---

> ### Author Response · Authors · 2025-11-27
> **Gentle Reminder**
>
> Dear Reviewer J5S5,
>
> Thank you again for taking the time to review our work. As the discussion phase will close in less than one week, to allow us sufficient time to address your comments, could you please take a moment to look over our responses and let us know if you have any remaining questions? Thank you very much!
>
> Best regards, Authors

---

### Official Review · Reviewer_Z6N3 · 2025-10-31

**Soundness:** 2
**Presentation:** 3
**Contribution:** 2
**Rating:** 4
**Confidence:** 3

**Summary:**

This paper proposes COCA, a data pre-processing framework designed to make safety reasoning explicit in fine-tuning data. COCA reformats training examples into a structured reasoning sequence using five tags, aiming to encourage the model to reason about and remove harmful content before producing final outputs. The authors provide theoretical analysis suggesting that such structured reasoning can concentrate harmful concepts into a more linearly separable subspace in the model’s internal representation space. Comprehensive experiments on several open-weight LLMs show that COCA enhances model's resistance to both in-distribution and out-of-distribution attacks while maintaining utility.

**Strengths:**

1. The proposed method is easy-to-integrate in practice. Since it requires no system changes and only modifies the training data, COCA could be easily applied to existing fine-tuning framework.
2. The paper provides a formal analysis demonstrating that COCA encourages concentration of harmful concepts into a linearly separable subspace.
3. The authors present visualization result to illustrate the better internal separation after fine-tuning with COCA's data.

**Weaknesses:**

1. The authors evaluate COCA primarily on medium-sized models (7B-9B). Including smaller (<=3B) and larger (>=14B) models would help assess COCA's generalizability.
2. The paper claims that COCA is orthogonal to SRG and STAIR and could be combined with them, yet no experiments demonstrate this. Moreover, COCA’s reported performance in Table 1 does not show clear improvement over these baselines.
3. COCA relies on five hardcoded tags. The paper does not clarify how these tags were defined or whether performance depends on their semantics or order. It remains unclear if changing, removing, or reordering tags would affect performance.
4. The theoretical analysis models COCA’s data as a generic structured reasoning sequence and does not differentiate the role of individual tags. As a result, the theory does not explain how specific tags contribute to concept concentration. It is unknown that whether the effectiveness of COCA comes from its five-tag design or only from its longer reasoning context provided during the fine-tuning.
5. COCA's main contribution is related to train the model to follow a specific reasoning pattern through fine-tuning. Therefore, comparisons with similar reasoning-based alignment methods, such as Deliberative Alignment [1], are necessary.
6. The paper presents internal-state separability results only for COCA, while ignoring the baselines. Without this comparison, it is unclear whether the observed separation is unique to COCA or a general outcome of fine-tuning.
7. COCA evaluates internal-state separability but does not use these states to classify or guide safety decisions. Higher separability in internal representations does not necessarily imply improved output safety and quality of the model.

[1] Guan, Melody Y., et al. "Deliberative alignment: Reasoning enables safer language models." arXiv preprint arXiv:2412.16339 (2024).

**Questions:**

Please check my questions in the Weaknesses section.

---

> ### Author Response · Authors · 2025-11-22
> **Response to Reviewer Z6N3 (Part 1)**
>
> Thank you for your time and effort in reviewing our work. Please find our responses to your questions and concerns below.
>
> > W1 Evaluation on smaller and larger models.
>
> We agree that including smaller and larger models could further help to assess COCA's generalizability. Hence, we have extended the evaluation to smaller and larger models. Concretely, we have added `Qwen2.5-3B` as a ≤3B model and `Qwen2.5-14B` as a ≥14B model. The training recipe and data mixture are kept identical to the 7B–9B setting with LoFiT as the intervention module. The results show that COCA is **generalizable to smaller and larger-sized models**.
>
> |        | PAIR | JChat | Cipher | Comp | Code | JailWild | Avg|
> |--------|-------------|-----------|--------|------|------|------------|------|
> | Instruct (Qwen-2.5-3B)  |       48.4   |     28.5  |  15.0   | 8.5  | 57.5  |    12.7    |       28.4  |
> | Base + COCA (Qwen-2.5-3B)  |  23.4        |  8.0     |  0.5   | 0.0  | 40.5 |     8.3   |    13.5     |
> | Instruct (Qwen-2.5-14B)   |    39.1      |   30.0    |   13.5  | 10.0  | 60.5 |      9.3   |    27.1  |
> | Base + COCA (Qwen-2.5-14B)   |   9.3       |    7.0   |  0.0   | 0.0  | 36.0 |   5.7      |    9.6  |
>
> > W2 Orthogonal to SRG and STAIR; improvement over the baselines
>
>
> We have implemented and evaluated a **combined variant that explicitly composes STAIR’s introspective stage with COCA’s concept-concentration protocol.** Concretely, we feed STAIR’s “Problem Analysis” into COCA’s `<think>` segment, then apply the COCA `<concept> → <check> → <erase unsafe concepts> → <response>` pipeline, and train the same LoFiT intervention on the resulting refactored traces. Here, we used STAIR's publicly released data. We further refactor the data under our COCA template and then combine them. The results are given below. When COCA and STAIR were combined, a **4.7% reduction in attack success rate** was further achieved on OOD jailbreak attacks.
>
>
>
> |        | PAIR | JChat | Cipher | Comp | Code | JailWild | Avg|
> |--------|-------------|-----------|--------|------|------|------------|------|
> | COCA  |    35.9      |  13.5     |   3.5  | 0.0   | 42.5  |     5.7   |   16.9      |
> | STAIR   |   31.3       |  18.0    |  3.0   |  0.0 | 40.5 |   6.7     |      16.6  |
> | COCA + STAIR    |    23.4      |   7.5    |   0.0  | 0.0  | 36.5  |    5.7     |   12.2   |
>
>
>
> STAIR achieves a strong number on PAIR when evaluated in isolation. However, when averaging across all OOD suites, COCA still provides a 1.1% reduction on attack success rate with LoFiT intervention and 4.5% reduction on attack success rate with RR intervention. Furthermore, STAIR adopted 3K Jailbreak prompts from Jailbreak-V as their training data. Our COCA was trained only on the illegal instructions without jailbreak prompts, which is a more challenging scenario to achieve OOD robustness.
>
> > W3 Clarification on how tags were defined or whether performance depends on their semantics or order.
>
> In COCA, the reasoning protocol is derived from the concept-concentration analysis.  In practice, we use four explicit control tags:`<think>, <concept>, <check>, and <erase unsafe concepts>`, followed by the terminal `<response>`. Each of the tags (`<think>, <concept>, <check>, <erase unsafe concepts>`) plays a definite role in **linearizing the harmful direction**:
>   - The `<concept>` stage forces the model to restructure an explicit, compressed representation of unsafe content.
>   - The `<check>` stage conditions downstream behaviour exclusively on that representation.
>   - The refusal logic is separated in `<erase unsafe concepts>`, ensuring that benign requests will not be bypassed.
>
>
> To further demonstrate the effectiveness of the tag design in COCA, we have conducted ablations to remove the `<concept>` related tags and keep only `<think>`. This produces lower safety robustness versus the full COCA design. Reordering the stages to apply `<check>` before `<concept>` is not consistent with the intuition that we can only conduct safety checks or editing reliably after concept identification.
>
> |        | PAIR | JChat | Cipher | Comp | Code | JailWild | Avg|
> |--------|-------------|-----------|--------|------|------|------------|------|
> | COCA  |      35.9    |   13.5    |   3.5  | 0.0  | 42.5  |    5.7    |    16.9     |
> | COCA (Concept Remove)   |      43.7    | 24.5   |  2.0   | 0.5  |50.5  |  13.1       | 22.4     |
>
> The empirical results also imply that the tag design is sensible and essential as Corollary 3.3 proves that, at any stationary point, covariance between the concentrated representation and the harmful concept collapses onto a single direction to concentrate the harmful concepts into a linear subspace. The `<concept>` stage is what induces this collapse, the `<check>` stage is what gates on it, and the `<erase unsafe concepts>` stage is what decouples refusal from benign helpfulness. Removing or reordering these breaks the assumptions required for concentration.

---

> ### Author Response · Authors · 2025-11-22
> **Response to Reviewer Z6N3 (Part 2)**
>
> > W4 Clarification on difference between COCA and only longer reasoning context
>
>
> As in the response to W3, we have conducted ablations to remove the `<concept>` related tags and keep only `<think>`. This produces lower safety robustness versus the full COCA design. While `<think>` tag already produces longer chains of thought,  generic “more reasoning” is not sufficient for safety generalization. The key is to linearize where harmful concepts reside. COCA provides the structural pressure to concentrate harmful concepts so that representation intervention can work faithfully.
>
> > W5 Comparison with Reasoning-based alignment
>
>
> We have already compared SRG and STAIR, both of which are the state-of-the-art reasoning-based safety supervision methods. COCA+LoFiT reduces OOD jailbreak success relative to SRG+LoFiT by 6.7% on average.
>
> For deliberative alignment [1], as the detailed template is not open-sourced, we didn’t compare it directly. Although o1-series models are publicly accessible through APIs, the API had frequent warnings of jailbreak attempts during API calls. The same behaviour was also noted in STAIR paper. So we did not include it as a comparison. In Table 3, DeepSeek-R1, whose reinforcement learning post-training might overlap some similarities and be related to Deliberative Alignment, is outperformed by our COCA method.
>
> > W6 The paper presents internal-state separability results only for COCA, while ignoring the baselines.
>
> We have added internal separability analyses for all baselines on Qwen-2.5-7B using LoFiT intervention. For each method, we collect hidden states at layer 16 for illegal, jailbreak, and benign prompts, respectively. We train a single linear probe to distinguish unsafe vs. benign, and report ROC-AUC.  We trained the probe on illegal and benign prompt internal states. We evaluate the AUC on jailbreak and benign prompt internal states. The results show that **COCA exhibits consistently higher separability than SRG+LoFiT, STAIR+LoFiT, and the vanilla baseline**. This demonstrates that the observed separation is not a generic byproduct of fine-tuning. Instead, **the theory-motivated tag design in COCA specifically sharpens the geometry along the harmful direction**.
>
> |        | Vanilla | COCA | STAIR | SRG | RR |
> |--------|---------|------|--------|------|----|
> | AUC  |       0.85 |  **0.96**    |    0.93 | 0.90  | 0.91 |
>
>
> > W7 Clarification on the high separability and improved output safety
>
>
> We need to clarify that our goal is not to deploy probes at inference for classification or detection. Instead, we aim to simplify the geometry of representation space and train the representation intervention module so that a simple editor (e.g., LoFiT) can act faithfully along the harmful direction identified by the concept-concentration protocol. This is precisely the linear representation hypothesis [2] underpinning representation interventions: if harmful content is concentrated into a low-dimensional subspace, a linear editor can reliably gate or erase it. The internal separability analysis thereby serves as a proof-of-concept: under COCA, the harmful direction becomes more linearly accessible. Crucially, the improvements we report in Tables 1 and 3 are **output-level safety metrics** (attack success rates and helpfulness), not probe scores. We have not used internal states' separability to classify or guide responses at inference.
>
> [1] Guan, Melody Y., et al. "Deliberative alignment: Reasoning enables safer language models." arXiv preprint arXiv:2412.16339 (2024).
>
> [2] Park K, Choe Y J, Veitch V. “The Linear Representation Hypothesis and the Geometry of Large Language Models.” International Conference on Machine Learning. PMLR, 2024: 39643-39666.

---

> ### Author Response · Authors · 2025-11-26
>
> Dear Reviewer Z6N3,
>
> We are grateful for your time and constructive feedback. To facilitate our discussion at this midway point, we provide a concise summary of your concerns and our responses:
>
> > W1 Model scale generalization (≤3B and ≥14B)
>
> We extended evaluations to Qwen2.5-3B and Qwen2.5-14B. COCA consistently reduced OOD attack success rates, indicating generalizability beyond 7B–9B.
>
> > W2 Orthogonality and composability with SRG/STAIR
>
> We implemented a combined variant (COCA+STAIR). This yielded a further 4.7% reduction in OOD ASR, supporting composability.
>
> > W3 & W4 Role and necessity of tags
>
> We add additional experiments and discussions on the tags design. Ablating concept-related tags degrades robustness (avg ASR 16.9 → 22.4). The design aligns with our theory (Corollary 2.3) on concentrating harmful concepts into a linear subspace.
>
> > W5 Comparisons with other reasoning-based alignment
>
> We compared against reasoning-based alignment SRG and STAIR. Direct comparison to Deliberative Alignment [1] is infeasible due to API safety filtering. However, COCA outperforms DeepSeek-R1 (Table 3), whose RL post-training may share similarities with deliberative approaches.
> > W6 Internal-state separability across baselines
>
> We added linear-probe analyses. COCA achieves the highest ROC-AUC (0.96), showing COCA sharpens harmful directions beyond generic fine-tuning effects.
>
> > W7 Using separability vs. output safety
>
> We provide a discussion on separability and output safety. Separability is used to validate that COCA makes the harmful direction more linearly accessible, enabling simple representation interventions to act faithfully. Our reported gains are output-level safety metrics, not probe scores.
>
> Thank you again for your thoughtful review. We would be grateful if you could review our clarifications and let us know if any questions remain or if further evidence would help. We sincerely appreciate your time and consideration.
>
> Best regards, Authors

---

> ### Author Response · Authors · 2025-11-27
> **Gentle Reminder**
>
> Dear Reviewer Z6N3,
>
> Thank you again for taking the time to review our work. As the discussion phase will close in less than one week, to allow us sufficient time to address your comments, could you please take a moment to look over our responses and let us know if you have any remaining questions? Thank you very much!
>
> Best regards,
> Authors

---

### Author Response · Authors · 2025-12-03
**Summary of Rebuttal (Part 1)**

## Summary of Rebuttal

Dear Reviewers, Area Chair, Senior Area Chair and Program Chair,

We appreciate the time and effort required to evaluate submissions under this year’s exceptional circumstances. To assist your assessment, we provide a concise overview structured as follows: (1) the paper’s core motivation and novelty, (2) a summary of the discussions, and (3) a summary of responses to the major concerns raised.


## Core Motivation and Novelty


**Concept Concentration (COCA):** A data-level linearizer that refines standard safety instruction data into a structured reasoning protocol to concentrate harmful concepts into a near-linear subspace, enabling robust and faithful safety alignment via concept-centric representation interventions. Core novelties of COCA include:

**(a) A Theory-Driven Shift from Complex Intervention to Data-Level Linearization ⇒** resolves the fundamental limitation where non-linear entanglement of harmful and benign concepts makes perfect erasure impossible (Theorem 2.2, Corollary 2.3). The paper `provides a formal analysis demonstrating that COCA encourages concentration of harmful concepts into a linearly separable subspace` (Reviewer Z6N3). This formal analysis provides a `clear theoretical contribution`, is `insightful and valuable to the safety community` (Reviewer 3MTC, eUou).

**(b) A Practical, Structured Linearization Method via Data-Level Refactoring  ⇒** resolves the fragility of applying representation interventions to standard models by enforcing a concept-identification-gating-erasure process via data refactoring. The method is `intuitive, easy to follow` (Reviewer eUou) and `easy-to-integrate in practice` as it requires no system changes (Reviewer Z6N3). This yields `strong empirical results` (Reviewer 3MTC) and `better internal separation` (Reviewer Z6N3).

## Outcomes of the Discussion

We are grateful to all reviewers for their detailed comments. We are encouraged to have the engaged reviewers and discussions, which we summarize the discussions as follows:

*   **Reviewer 3MTC (Rating: 8):**
    *   Recognized the clear theoretical contribution, novel methodoglogy and strong empirical results.
    *   Initially raised concerns about annotator bias and evaluation on Large Reasoning Models (LRMs). After we provided new experiments showing consistent performance across annotators (GPT-4o, Claude, LLaMA-3.1-8B) and results on DeepSeek-R1 versus LRM baselines (SAFEPATH, Star-1), the reviewer explicitly confirmed on **November 23, 2025**, that `these additions directly addressed the concerns` and that the `now confident that it deserves acceptance.` **(raised the confidence score to 5)**.  This confirmed high score and acceptance stance was provided **prior to the reported OpenReview bug**.
*   **Reviewer eUou (Rating: 6):**
    *   Recognized the paper's important problem and insightful theory.
    *   Concerns about table presentation, ID/OOD definition, and lack of gradient-based attack evaluation were addressed by restructuring tables, adding clarifications, and including new GCG attack results showing strong robustness.
*   **Reviewer J5S5 (Rating: 2):**
    *   Acknowledged the problem is valuable and the paper is well-written.
    *   The core concern is that the method was `just like prompt or CoT engineering`. We provided **direct theory-driven justifications, ablation studies** proving the necessity of the structured tags, and a **prompt-only baseline** showing **it fails (30.9% ASR) versus COCA fine-tuning (10.5% ASR)**, demonstrating COCA is not a simple prompting trick.
*   **Reviewer Z6N3 (Rating: 4):**
    *   Acknowledged the method's practical design and theoretical analysis.
    *   Key concerns on generalizability, composability, tag design, and internal-state comparisons were resolved with new experiments on 3B/14B models, a combined COCA+STAIR variant, tag ablation studies, and added linear probe results across all baselines showing COCA's superior separability (0.96 AUC).




## Point-by-Point Response to Major Concerns


We have provided direct responses to all reviewers' concerns with new experiments, analyses, and clarifications, as summarized below.

### Extensions and Comparsions with other Deliberative Reasoning Methods (Z6N3, eUou, 3MTC)

> Concern: Orthogonality and composability with state-of-the-art methods like STAIR. `(Reviewer Z6N3 W2, Reviewer eUou W2)`

**Response:** We implemented and evaluated a combined COCA+STAIR variant. This hybrid approach **achieved a further 4.7% reduction** in average OOD ASR compared to COCA alone (16.9%→12.2%), demonstrating clear orthogonality and practical synergy.

> Concern: Need for comparison with other reasoning-based alignment methods. `(Reviewer Z6N3 W5)`

---

> ### Author Response · Authors · 2025-12-03
> **Summary of Rebuttal (Part 2)**
>
> **Response:** We have compared with SOTA reasoning-based methods SRG and STAIR, with COCA showing an **average 6.7% reduction in OOD ASR over SRG+LoFiT**. A direct comparison to Deliberative Alignment was infeasible due to API restrictions on jailbreak prompts. In our evaluations, COCA outperformed DeepSeek-R1, a model whose reinforcement learning training shares conceptual similarities with deliberative reasoning.
>
>
> > Concern: Evaluation on Large Reasoning Models (LRMs). `(Reviewer 3MTC W2)`
>
> **Response:** We applied COCA to DeepSeek‑R1‑Qwen‑7B and compared it to recent LRM safety baselines (SAFEPATH, Star-1). COCA achieved competitive or lower average OOD ASR (11.7%) versus these baselines (13.1%, 14.4%), confirming its applicability and effectiveness on reasoning-strong models.
>
> **Feedback**: Reviewer 3MTC kindly acknowledged our responses addressed the concerns.
>
> ### Distinguishing from Prompting-only Methods (J5S5, Z6N3)
>
> > Concern: Whether COCA’s benefits stem merely from prompting rather than concept concentration. `(Reviewer J5S5 W1&W2)`
>
> **Response:** We ran a new **prompt-only baseline** where the COCA template was prepended at inference *without any fine-tuning*. **Its performance was substantially worse (30.9% avg OOD ASR) compared to COCA with supervised fine-tuning (10.5% ASR).** This critical experiment demonstrates that the supervised concentration step is necessary; COCA is not a simple prompting trick. We also provided theory-driven justifications, ablation studies proving the necessity of the structured tags and the improved internal state separability.
>
> > Concern: Necessity and justification of the tag design in COCA. `(Reviewers Z6N3 W3&W4, J5S5 W1)`
>
> **Response:** We conducted an ablation study removing the core `<concept>`-related tags, resulting in a generic “think-only” chain-of-thought. This **significantly degraded OOD robustness**, increasing the average ASR from 16.9% to 22.4%. This empirical result proves the tag design is not arbitrary but is essential for the concept concentration mechanism.
>
>
>
> ### Analysis of Concept Concentration for Improved Separability and Safety (Z6N3)
>
> > Concern: Clarification on the relationship between internal-state separability and improved output safety. `(Reviewer Z6N3 W7)`
>
> **Response:** We clarified that the goal is not to use probes at inference. Higher separability validates that COCA makes harmful concepts more linearly accessible, enabling representation interventions (e.g., LoFiT) to act faithfully. Crucially, all improvements reported (e.g., in Tables 1 & 3) are **output-level safety metrics**, confirming that the improved geometry translates directly to safer model outputs.
>
>
> > Concern: Comparison on internal-state separability across baselines. `(Reviewer Z6N3 W6)`
>
> **Response:** We added linear probe analyses for all baselines on Qwen2.5-7B. **COCA achieved the highest separability** (AUC: 0.96), outperforming STAIR (0.93), SRG (0.90), and a vanilla baseline (0.85). This shows COCA sharpens the harmful direction in representation space.
>
>
> ### Ablation Studies on other Model Scales, Gradient-based Attacks, and Annotators (Z6N3, eUou, 3MTC)
>
> > Concern: Generalizability of COCA to smaller and larger model scales. `(Reviewer Z6N3 W1)`
>
> **Response:** We added evaluations on Qwen2.5-3B (≤3B) and Qwen2.5-14B (≥14B). COCA **consistently reduced the average out-of-distribution (OOD) attack success rate (ASR) on these models** (e.g., 28.4%→13.5% for 3B, 27.1%→9.6% for 14B), confirming its effectiveness beyond the initially reported 7B-9B scale.
>
>
> > Concern:  Need for evaluation against gradient-based jailbreak attacks. `(Reviewer eUou W4)`
>
> **Response:** We added evaluation against the GCG attack. COCA significantly improved robustness, reducing ASR from 35.0% to 4.0% with LoFiT intervention and from 45.5% to 9.5% with ReFT intervention.
>
>
> > Concern:  Potential annotator bias in defining harmful concepts. `(Reviewer 3MTC W1)`
>
> **Response:** We evaluated COCA using training data generated by three different annotators (GPT-4o, Claude, LLaMA-3.1-8B). **All achieved similarly strong and low OOD ASR** (between 10.5% and 13.2%), demonstrating the method’s robustness to annotator variation.
>
>
> ### Clariticaion on Definitions (eUou)
>
> > Concern:  Readability of tables and clarification of ID/OOD definitions. `(Reviewer eUou W1&W3)`
>
> **Response:** We restructured Tables 1 and 2 to group methods by training paradigm to improve readability. We also clarified the ID/OOD definitions (ID: direct illegal instructions; OOD: jailbreaks with distribution shifts like role-play or ciphers).
>
>
> ---
>
> In summary, we believe the paper presents a theoretically grounded, practical method that addresses a core challenge in safety alignment. The reviewers' constructive concerns have been addressed with substantial new evidence, strengthening the paper's contribution. Again we are grateful for the reviewers' time and insightful feedback.
>
> Best Regards, Authors

---

### Meta-Review · Area_Chair_pAFh · 2026-01-07

**Summary:**

The paper proposes COCA to solve the non-linear entanglement problem in safety alignment by refactoring training data. Reviewers agreed that this work introduced an easy-to-integrated method with training data restructure.  For this paper, 4 reviewers have submitted their reviews. Some reviewer shows strong concerns about the incremental technical contributions. Others brought up concerns about the generalizability to different model sizes and the design rationale behind the specific tag design. There were also some concerns about the marginal improvement over state-of-the-art baselines. The theorem result is cool but could benefit from more discussions around "theoretically impossible" and "practically insufficient" for existing erase approaches.  The authors have made efforts to address the concerns raised by reviewers. However, the paper still suffers from limitations such as limited technical contributions and marginal empirical gains in some settings.

**Reviewer Concerns:**

For Reviewer J5S5, the prompt-only baseline addresses part of the skepticism, but the main contribution and novelty criticism remains outstanding and the rebuttal is unlikely to convince the reviewer. For Reviewer Z6N3, several experimental requests are addressed, but the core novelty concern regarding tag design remains outstanding. For Reviewer eUou, the gradient-based attack concern is addressed by the added GCG results, but suite-level advantages over STAIR remain not fully convincing. For Reviewer 3MTC, the major concerns are largely addressed by cross-annotator and LRM experiments.

**Reviewer Scores:**

No reviewers would change their score.

---

### Decision · Program_Chairs · 2026-01-26

Reject